# An Unexpected 12.6 Centimeter Nail in the Thorax Damaging Vital Structures: A Case Report "Nailed It"

**Thomas H. Avedissian** [1,*]**, Daniel J. F. M. Thuijs** [2]**, Lucas Timmermans** [3]**, Alexander P. W. M. Maat** [2] 
**and Edris A. F. Mahtab** [2,*]

1    Department of Emergency Medicine, Albert Schweitzer Hospital, Albert Schweitzerplaats 25,
    3318 AT Dordrecht, The Netherlands
2    Department of Cardiothoracic Surgery, Erasmus University Medical Center, Dr. Molewaterplein 40,
    3000 CA Rotterdam, The Netherlands
3    Department of Trauma Surgery, Erasmus University Medical Center, Dr. Molewaterplein 40,
    3000 CA Rotterdam, The Netherlands
*    Correspondence: thomasavedissian@hotmail.com (T.H.A.); e.mahtab@erasmusmc.nl (E.A.F.M.)

**Abstract:** We report a patient who was referred to the emergency room with pulmonary complaints and where a computed tomography (CT) scan showed an unexpected 12.6 cm nail in the thorax penetrating part of the left pulmonary upper lobe, the left pulmonary artery, the left main bronchus, and the descending aorta, which had been in situ for at least three days. The quickly deteriorating patient had to be transferred to a tertiary academic hospital where the nail was successfully surgically removed. The comprehensive description of this unique case with a discussion of the critical decision moments could render insights into the management of challenging trauma cases.

**Keywords:** thoracic trauma; penetrating injury; foreign body; thoracic surgery; radiology

## 1. Introduction

Thoracic trauma is one of the main causes of death in trauma patients and is often seen in emergency departments [1]. Penetrating chest injuries account for 1–13% of thoracic trauma hospital admissions [2]. Mortality rises as complications such as tamponade, haemothorax, tension pneumothorax, or tracheobronchial injury are present [3]. Primary resuscitation methods and surgical management may differ per case depending on the mechanisms of injury and vital structures that are compromised. Remarkable challenges in patient management are imposed in those cases when a foreign body is still present in the thorax after the trauma [4,5].

Here we report a case of a patient who presented with a 12.6 centimetre (cm) long nail in his thorax, after being referred to the emergency room with atypical pulmonary complaints. The descriptions and images of the initial work-up and surgical procedure could render insights in the management of such challenging thoracic trauma cases.

## 2. Case Presentation

A healthy, 32-year-old man was referred to the emergency department of a local hospital without thoracic surgical capabilities. The patient reported complaints of fever, dyspnea, chest pain during respiration, and a productive cough containing small amounts of blood for three days. The referring primary care physician suspected pneumonia, as the patient had suffered this before. The patient explained that he recognized his current complaints from previous episodes of pneumonia and reported no history of trauma to the chest or other significant events.

During the evaluation at the emergency department, the vital signs of the patient were as follows: respiratory rate 36 times per minute, oxygen saturation 99% without supplemental oxygen, heart rate of 102 beats per minute, blood pressure of 136/80 mmHg,

and a temperature of 38.5 degrees Celsius (101.3 degrees Fahrenheit). Upon physical examination it was noticed that respirations were superficial, symmetrical, and appeared painful, with crepitus over the lower quadrant of the left lung on auscultation. The patient insisted on sitting in an upright position for comfort. No external abnormalities were seen on the thorax at first examination. Laboratory results showed a white count of $9.9 \times 109/L$ (reference: $4.0–10 \times 109/L$) and a C-reactive protein (CRP) of 157 mg/L (reference < 5.0 mg/L). A rapid multiple polymerase chain reaction (PCR) for SARS-CoV-2 from nasopharyngeal swabs was positive. Because a pulmonary embolus was suspected to accompany a pulmonary infection, a computed tomographic angiography (CTa) of the thorax was performed without first making a chest X-ray (Figure 1, Videos S1 and S2).

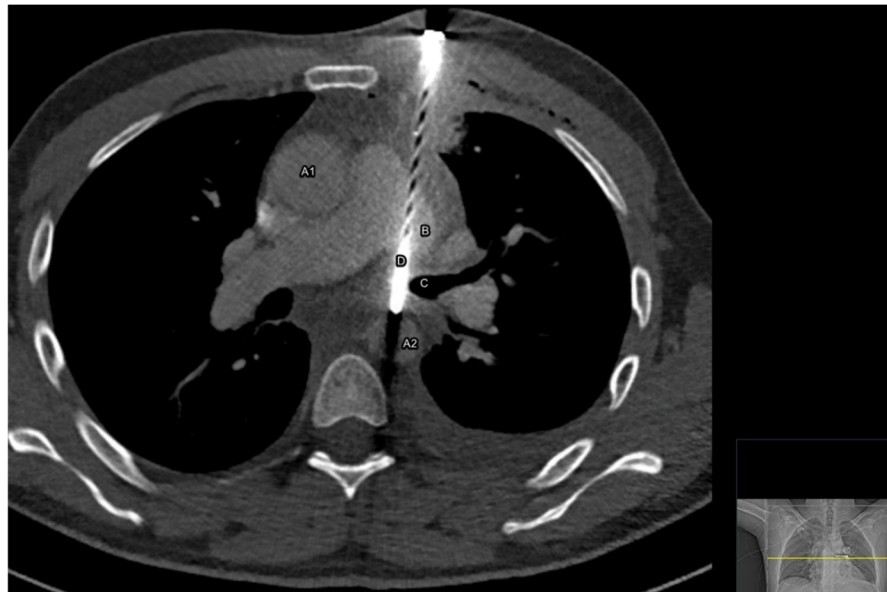

**Figure 1.** Transverse image of the chest CT scan showing a 12.6 cm long nail and its trajectory through the thorax and vital structures. A1 = ascending aorta, A2 = descending aorta, B = left pulmonary artery, C = left main bronchus, and D = nail (12.6 cm).

The CT scan showed an unexpected foreign body with the radiographic appearance of a nail of 12.6 cm, with the entrance at the left parasternal between costa two and three, penetrating part of the left pulmonary upper lobe, the left pulmonary artery, and the left main bronchus and with its tip positioned in the descending aorta. Small amounts of fluid were visible along the ascending aorta, in the pericardial sac and in the left pleural space, accompanied by a small pneumothorax at the top of the left lung.

Upon re-examination of the patient, a small, circular, dry wound with a palpable pulsating hard object underneath was discovered on the thorax (Figure 2 and Video S3), which was not noticed before because it was hidden by an electrocardiogram (ECG) sticker during the first examination. The patient refused to tell how or when the object had been fired into his chest. After contacting the place of residence of the patient, the care team was told that the patient had worked with a nail gun three days earlier, and was noticed to have a depressed mood the last few weeks. However, it was not noticed or suspected that the nail gun had been used by the patient himself or somebody else.

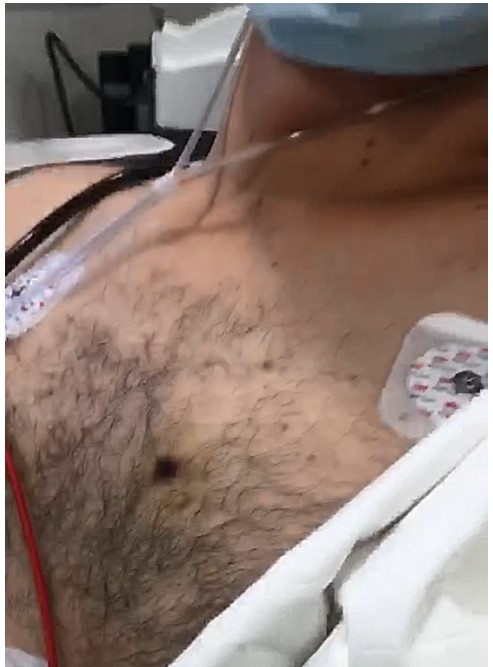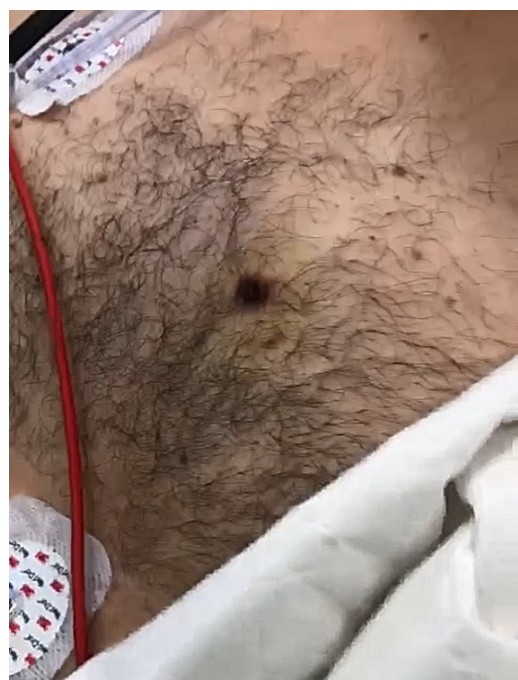

**Figure 2.** Puncture wound with hematoma at the 4th intercostal space of the left hemithorax. During the first examination in the emergency room the wound was hidden behind an ECG sticker.

The patient's condition had worsened after he had to lay in the supine position for the CT scan. He now actively started to cough up small amounts of blood and needed 15 L of supplemental oxygen to maintain an oxygen saturation of above 94%. His heart rate elevated to 160 beats per minute and his blood pressure dropped to 120/100 mmHg. It was suspected that manipulation of the nail following repositioning of the body caused the deterioration. A thoracic surgeon from a tertiary hospital was consulted and priority in management was given to transferring the patient to the nearest hospital with cardiothoracic surgical capabilities as quickly as possible, while limiting movements of his upper body. After consideration, it was chosen not to place a chest tube or to perform endotracheal intubation in the emergency room because this would delay transport and risk further manipulation of the nail. The patient was given crystalloid fluids, packed red blood cells, tranexamic acid, antibiotics, tetanus toxoid, opiates, and anti-emetics because these interventions did not give a delay and could be continued during transport. He was moved to the nearest tertiary care centre via ambulance under the supervision of a trauma surgeon of the Mobile Medical Team (MMT). The total duration of transport was 17 min.

*Operative Findings*

At presentation at the Emergency Room of the Erasmus University Medical Centre (Erasmus MC), a tertiary university trauma centre, the patient was tachycardiac, minimally conscious, and in acute respiratory distress. A swift transfer to the operating theatre was initiated and the patient was prepared for immediate surgery. Intubation with a double-lumen tube was performed in order to achieve single-lung ventilation.

The surgical strategy to remove the nail was determined by the preoperative CT scan of the thorax. The 12.6 cm nail entered the thorax at the left 2nd intercostal space, pierced through a segment of the left upper lobe of the lung, and went through the left main pulmonary artery as well as the left main pulmonary bronchus. The distal tip of the nail punctured the descending thoracic aorta. A staged extraction of the nail, starting at the descending aorta and proceeding with the other vital structures as mentioned above, was aimed for.

The patient was positioned in a right semi-lateral recumbent position with 30 degrees of thorax rotation to the right, and an almost supine position of the pelvis to facilitate

femoral cannulation to use extra corporeal circulation (ECC). A left anterolateral thoracotomy was performed at the level of the fourth intercostal space.

First, two wedge excisions of the left upper lobe, surrounding the nail, were performed using a short endoGIA stapler. We proceeded with the left main bronchus, where the nail pierced through and through. No purse string sutures were placed at this point, as we decided that we would reconstruct the left main bronchus after the nail was completely removed.

Then, a ventral and dorsal 4.0 polypropylene purse-string suture, around the puncture site of the nail, was placed in the left main pulmonary artery. In order to minimize pulmonary blood circulation during the manipulation of this delicate structure (pulmonary artery) when placing the purse-string sutures and later removal of the nail, ECC was required. This furthermore enabled rapid auto-transfusion when excessive blood loss was encountered.

Heparin (25,000 units) was administered to achieve an adequate activated clotting time (ACT > 450 s) and ECC was initiated via the left common femoral artery (19 French coated cannula) and vein (25 French coated cannula), without the need for antegrade arterial cannulation in the left superficial femoral artery.

Then, the pericardium was opened to achieve surgical control over the main vessels during the subsequent steps of the surgical procedure. The pulmonary veins were intact. Finally, the dorsal pleura was opened where a contained hematoma surrounding the descending thoracic aorta was present, without active bleeding. The descending aorta was mobilized without damaging the side branches. At the descending aorta, at the ventral side, a 4.0 pledgeted polypropylene purse-string suture was placed. A side clamp was placed on the aorta around the nail. Now, all preparations to remove the nail were established and the staged removal of the nail was initiated by removing the nail from the descending aorta, the left main bronchus, and finally the left main pulmonary artery (Figure 3).

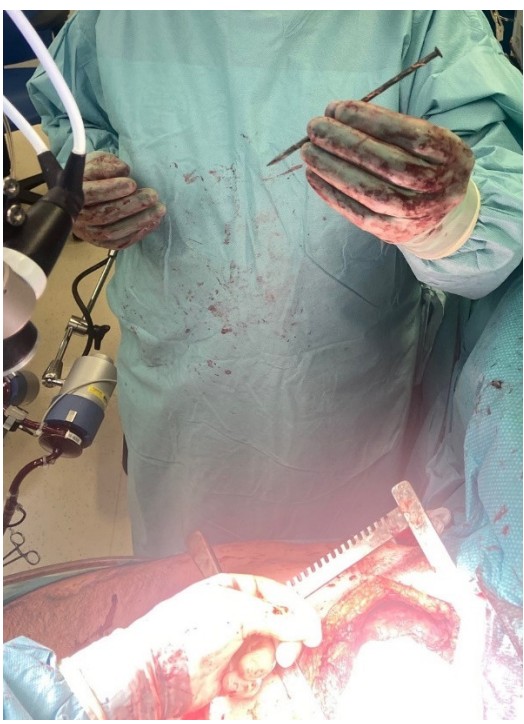

**Figure 3.** The 12.6 cm nail after surgical removal.

The purse-string sutures were tightened and the ventral and dorsal defects of the left main bronchus were closed with a polydioxanone (PDS) suture. Bronchoscopy was performed during surgery to aspirate blood clots and sputum to ensure adequate ventilation and check for an open ostium of the left main pulmonary bronchus. The patient was

successfully weaned from ECC. After haemostasis and adequate insufflation of the left lung, the thoracic cavity was closed over three chest drains (one in the pericardium, one in the pleural sinus, and one in the upper thoracic aperture). The patient was transferred in stable cardiopulmonary condition to the Intensive Care Unit (ICU).

The postoperative period was without complications and the patient recovered satisfactorily. Prophylactic intravenous ceftriaxone antibiotic treatment was administered for one week. The chest tubes were removed at the third postoperative day and the patient was transferred to a secondary care centre on the sixth postoperative day (Figure 4).

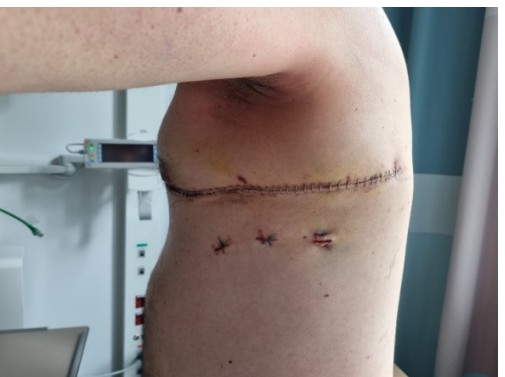 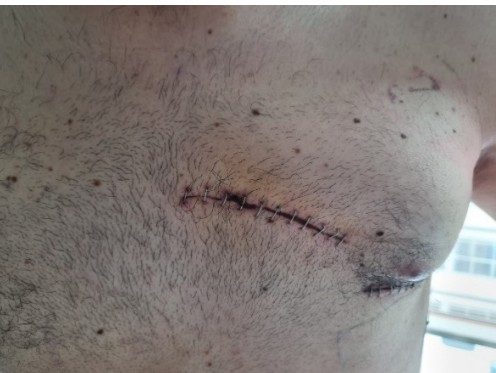

**Figure 4.** Antero-lateral incision with three chest tube puncture sites (left image). The right image depicts the incision where the nail was removed from the 4th intercostal space of the left hemi-thorax.

## 3. Discussion

Although thoracic trauma is a common presentation in the emergency room, frequently leading to hospital admission, the presence of foreign bodies in the thorax after trauma remains rare [1]. Despite multiple published studies describing management of cases where thoracic foreign bodies are present after trauma, there is no universal protocol or treatment plan due to the heterogeneity in clinical presentation of patients and vital structures that are damaged [6].

Here, we report a case where the emergency care team encountered a patient who was referred with pulmonary complaints and unexpectedly had a large nail in his thorax and subsequently deteriorated rapidly. Treatment decisions had to be made by a team in a hospital that did not frequently encounter severe thoracic trauma cases. Much can be learned from analysing such cases to improve future decision making.

The first important aspect in the course of this case was that the nail in the thorax was not noticed during the initial physical examination by the referring primary care giver and the emergency physician. The main reason the nail was missed, was because it was not expected to be present in a patient complaining of coughing and a fever and leaving out any traumatic events to the chest. The positive PCR-test for SARS-CoV-2 increased the suspicion of a pulmonary infection as cause of the symptoms instead of a traumatic one. Moreover, the patient actively tried to hide the nail's entry wound on his chest by moving the ECG sticker to cover it up in the emergency room. However, a thorough and complete examination of the thorax including inspection and palpation of the chest would have detected the entry wound with the underlying nail at an earlier stage [7]. The result would have been an earlier consultation of a surgeon in a thoracic surgical centre, as well as trying to reduce the chance of manipulating the nail while obtaining CT-images of the chest. This shows the importance of a comprehensive physical examination of the chest for patients presenting with pulmonary complaints and to keep an open mind for differential diagnoses [8].

Another critical moment in this case was when the patient's condition started to deteriorate rapidly after the CT scan and the nail in the thorax was discovered. It was clear that the patient needed surgery, which also meant he needed to be transferred to a hospital with cardiothoracic surgical capabilities. At this point, the decision had to be made

to either make the transfer happen as soon as possible, or to make efforts to stabilize the patient before transferring him. This dilemma is known as 'scoop and run vs. stay and play (or treat then transfer)' [9]. It could be argued whether there was an indication to place a chest tube to treat the hemopneumothorax and to perform endotracheal intubation in the emergency room to try to optimize ventilation before transport [10,11]. However, in this case, priority was given to transfer the patient as quickly as possible for surgical treatment while keeping upper body movements to a minimum so as to reduce the chance of further manipulation of the nail. If needed, the trauma surgeon of the MMT on the ambulance could still perform these interventions during transport, although the working conditions there are worse. The interventions that were performed in the emergency room were those that did not give any delay and could be continued during transfer. Crystalloid fluids and packed red blood cells were given to treat haemorrhagic shock, as well as tranexamic acid to improve coagulation [12,13]. Additionally, opiates and anti-emetics were given for patient comfort, making it easier for him to minimize his upper body movements during transport [14]. Lastly, antibiotics and tetanus toxoid were given to treat any ongoing infections caused by the nail and as prophylaxis for potential future infections [15,16].

Finally, each thoracic trauma patient is unique due to the cause of trauma and the way it was inflicted, including sharp versus blunt, high energetic trauma (HET) versus compression, or with versus without foreign bodies remaining in the thoracic cavity. All these varieties are steering the optimal surgical approach to stabilize and treat the trauma patient. The surgical approach should be tailor-made based on the individual patient injuries, keeping in mind the main goal: "treat first what kills first" [11,17]. Furthermore, the need of consulting a (cardio)thoracic surgeon should be carefully and timely made by the emergency physician, as the (cardio)thoracic surgeon is not part of the standard emergency room trauma team and many hospitals do not have facilities for cardiothoracic surgery.

Especially for the patient described in the current case report, the surgical approach used was determined in order to be able to manage trauma to all vital structures involved, from retrosternal to the anterior wall of the descending aorta. This approach was based on the images of the CT scan in combination with bronchoscopic findings and the vital parameters of the patient. Adequate positioning of the patient on the operating table (a right semi-lateral recumbent position with 30 degrees of thorax rotation to the right and an almost supine position of the pelvis) is therefore crucial to retain the possibility of starting ECC through peripheral cannulation of the femoral artery and vein while starting with an anterolateral thoracotomy. The indication of whether to use ECC in individual trauma patients should be made with care, as it might benefit survival but it also could lead to increased morbidity. It is well known that the use of ECC can accommodate an inflammatory response that results in coagulopathy and temporary organ dysfunction affecting nearly every organ system [18]. Moreover, the need of complete heparinization of trauma patients could induce severe and life-threatening haemorrhage of organs and vital structures also compromised by the trauma. Here, we determined that ECC use was needed to obtain adequate venous drainage in order to minimize blood circulating through the right heart towards the lungs (e.g., passing through the left main pulmonary artery) to assure that the nail could be removed, in a controlled setting, from the left main pulmonary artery. The fruitful multidisciplinary collaboration in this specific patient ensured successful surgical removal of the 12.6 cm long nail and repair all compromised vital structures. The patient made an uncomplicated recovery and was transferred to a secondary care centre on the sixth postoperative day.

Several aspects made this case unique. First, the aspecific presentation at the emergency room of the referring secondary hospital and the unexpected finding of a 12.6 cm long nail on the CT images. Secondly, the nail was in situ for at least 3 days, without immediately comprising the patient's vital functions directly at the moment the nail was shot into his chest. Third, the tailor-made surgical approach was unique due to the need for specific positioning of the patient to ensure the possibility of femoral cannulation for ECC. Furthermore, the complete heparinisation of this trauma patient did not lead to detrimental

bleeding during surgical removal. Finally, the patient made a rapid and uncomplicated recovery while the nail could have led to life-threatening comorbidities and mortality.

## 4. Conclusions

Penetrating chest trauma with retained foreign bodies can be morbid and mortal and often require emergency surgical intervention. Each chest trauma patient presenting at the emergency room is unique due to the varying trauma mechanisms (sharp versus blunt, HET versus compression, and with versus without foreign bodies remaining in the thorax) which can result in a vitally stable versus unstable patient. Therefore, after initial resuscitation and management based upon protocols from Advanced Trauma Life Support (ATLS), a tailor-made surgical approach should be made for each chest trauma patient, weighing the pros and cons of each surgical step and possible use of extra corporeal circulation.

**Supplementary Materials:** The following supporting information can be downloaded at: https://www.mdpi.com/article/10.3390/surgeries4010006/s1, Video S1. CTa scan in axial view from cranial to caudal in venous phase, showing the nail and its relation to other structures in the thorax; Video S2. CTa scan in axial view from cranial to caudal in lung setting, showing the nail and its relation to other structures in the thorax; Video S3. Pulsating hematoma at the 4th intercostal space of the left hemithorax. One can notice the nail is pulsating with every heartbeat due to its trajectory and the tip barely touching the descending aorta.

**Author Contributions:** Conceptualization, T.H.A. and D.J.F.M.T.; methodology, T.H.A. and D.J.F.M.T.; resources, T.H.A. and D.J.F.M.T.; data curation, T.H.A. and D.J.F.M.T.; writing—original draft preparation, T.H.A. and D.J.F.M.T.; writing—review and editing, T.H.A., D.J.F.M.T., L.T., A.P.W.M.M. and E.A.F.M.; visualization, T.H.A. and D.J.F.M.T.; supervision, E.A.F.M. All authors have read and agreed to the published version of the manuscript.

**Funding:** This research received no external funding.

**Institutional Review Board Statement:** Not applicable.

**Informed Consent Statement:** Written informed consent was obtained from the patient to use CT scan images, photos, videos, and to publish this paper.

**Data Availability Statement:** Data are contained within the article.

**Conflicts of Interest:** The authors declare no conflict of interest.

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
