# Peer review of "An Unexpected 12.6 Centimeter Nail in the Thorax Damaging Vital Structures: A Case Report “Nailed It”"

_2673-4095, doi:10.3390/surgeries4010006_

Round 1

Reviewer 1 Report

This report describes a case of a patient, brought to the emergency room with vague thorax complaints but treated in a tertiary academic hospital for the removal of an undiscovered large nail. Although the case is presented with a 'high level of suspense' the overall message is clear and the paper could contribute to the general awareness in the management of trauma patients. And as such fits within the scope of the special issue of Surgeries. The subtitle 'nailed it' attracts attention but could be more integrated.

Author Response

Dear reviewer,

Thank you for your review and your feedback!

After reading the comments made by you and the other reviewers, we’ve made some minor changes to our manuscript. They can be seen in red in the file attached, as the ‘track changes’ function in Word was used.

Kind regards,

Dr. Thomas H. Avedissian, MD

Also on behalf of the co-authors: Daniel J.F.M. Thuijs, Edris A.F. Mahtab, Alexander P.W.M. Maat and Lucas Timmermans

Reviewer 2 Report

This case report is interesting and helpful for emergency physicians to deal with those patients with similar complaints and symptoms. It is worthy of publication.

Author Response

(The authors gave the same response as above.)

Reviewer 3 Report

The authors reported a patient with 12.6 centimeter nail in his thorax penetrating part of the vital organs in the manuscript entitled “an unexpected 12.6 centimeter nail in the thorax damaging vital structures: a case report”. Several concerns have been raised.

1. This is quite unique. It is possible to attempt to explain why vital sing was relatively preserved in this case?

2. Could you explain a little bit more about what was unique points in this surgery? 

Author Response

Dear reviewer,

Thank you for your review and your feedback!

Concerning your first comment: We also found it quite remarkable that the patient’s vital signs were relatively preserved upon first presentation, especially considering that the nail had probably been in his chest for 3 days. We think this was the case because the nail went through and through most vital structures (left pulmonary artery and left main bronchus). Because of this, the nail tamponaded the bleeding that followed after the structures had been damaged. The tip of the nail was placed against the ascending aortic wall. During the surgery, a contained hematoma was seen without active bleeding, which makes us think that upon the moment the nail went into the body it probably damaged the aortic wall without fully tearing it open, because that would have been a quick certain death. Instead, the bleeding seemed to contain itself.

Upon first presentation in the emergency room, the patient reported that he was getting dyspneic and feverish over the course of two days. His vital signs then were as follows: respiratory rate 36 times per minute, oxygen saturation 99% without supplemental oxygen, heart rate of 102 beats per minute, blood pressure of 136/80 mmHg and a temperature of 38.5 degrees Celsius. We believe these vital signs were a result of painful respirations and an ongoing infection with systemic inflammatory response, both inflicted by the nail.

We think the nail was slightly moved when the patient had to lay in a supine position for the CT-scan, because he started to deteriorate quickly right afterwards. He started to frequently cough up blood and his vital signs evolved to a oxygen saturation of 94% with 15 liters of supplemental oxygen, a heartrate of 160/min and a slight drop in blood pressure to 120/100. We think the slight movement of the nail led to new small bleedings in the left main bronchus and the upper lobe of the lung, explaining the respiratory decline and the coughing up blood. We don’t think that there was a massive bleeding from the pulmonary artery or the aorta at this point, because the blood pressure stayed relatively high.

Concerning your second comment: The tailor-made surgical approach was unique due to the need for specific positioning the patient to ensure the possibility of femoral cannulation for ECC. Furthermore, the complete heparinisation of this trauma patient did not lead to detrimental bleeding during surgical removal. And finally, the patient made a rapid and uncomplicated recovery while the nail could have led to life-threatening comorbidities and mortality. 

We have made some adjustments to our manuscript based on your comments. They can be seen in red in the file attached, as the ‘track changes’ function in Word was used.

Kind regards,

Dr. Thomas H. Avedissian, MD

Also on behalf of the co-authors: Daniel J.F.M. Thuijs, Edris A.F. Mahtab, Alexander P.W.M. Maat and Lucas Timmermans
